# Optimizing the Structure and Optical Properties of Lanthanum Aluminate Perovskite through Nb^5+^ Doping

**DOI:** 10.3390/nano14070608

**Published:** 2024-03-29

**Authors:** Wei Liu, Yang Zou, Yuang Chen, Zijian Lei, Lili Zhao, Lixin Song

**Affiliations:** 1Key Laboratory of Inorganic Coating Materials CAS, Shanghai Institute of Ceramics, Chinese Academy of Sciences, Shanghai 200050, China; 1000511440@smail.shnu.edu.cn (W.L.); zouyang@mail.sic.ac.cn (Y.Z.); chenya2022@shanghaitach.edu.cn (Y.C.); leizijian27@163.com (Z.L.); lxsong@mail.sic.ac.cn (L.S.); 2School of Chemistry and Materials Science, Shanghai Normal University (SNU), Shanghai 200234, China; 3School of Physical Science and Technology, ShanghaiTech University (STU), Shanghai 201210, China; 4School of Materials Science and Engineering, Shanghai University (SHU), Shanghai 200444, China

**Keywords:** high reflectivity, Nb^5+^ doping, oxygen vacancies, defects

## Abstract

This work involves the introduction of niobium oxide into lanthanum aluminate (LaAlO_3_) via a conventional solid-state reaction technique to yield LaAlO_3_:Nb (LaNb*_x_*Al_1−*x*_O_3+δ_) samples with Nb^5+^ doping levels ranging from 0.00 to 0.25 mol%. This study presents a comprehensive investigation of the effects of niobium doping on the phase evolution, defect control, and reflectance of LaNb*_x_*Al_1−*x*_O_3+δ_ powder. Powder X-ray diffraction (XRD) analysis confirms the perovskite structure in all powders, and XRD and transmission electron microscopy (TEM) reveal successful doping of Nb^5+^ into LaNb*_x_*Al_1−*x*_O_3+δ_. The surface morphology was analyzed by scanning electron microscopy (SEM), and the results show that increasing the doping concentration of niobium leads to fewer microstructural defects. Oxygen vacancy defects in different compositions are analyzed at 300 K, and as the doping level increases, a clear trend of defect reduction is observed. Notably, LaNb*_x_*Al_1−_*_x_*O_3+δ_ with 0.15 mol% Nb^5+^ exhibits excellent reflectance properties, with a maximum infrared reflectance of 99.7%. This study shows that LaNb*_x_*Al_1−*x*_O_3+δ_ powder materials have wide application potential in the field of high reflectivity coating materials due to their extremely low microstructural defects and oxygen vacancy defects.

## 1. Introduction

Rare earth LaAlO_3_ with an ABO_3_ crystal structure is a material that has attracted a lot of attention in recent years due to its unique properties. Its high melting point [1,2,3], chemical stability [4,5], and large band gap (3.8 eV) [6,7,8] make it a promising material for various applications. If the radius of the doping ion is close to La^3+^ or Al^3+^, LaAlO_3_, an essential cubic perovskite oxide, can be doped with significant concentrations of ions through substitution [9,10,11,12]. Due to its strong optoelectronic capabilities, the B site held by Al^3+^ can be replaced by transition metal ions to produce a variety of applications, including photodetector [13,14] and luminescent phosphor [15,16], which have been reported in numerous papers. Furthermore, due to its distinctively extended fluorescence lifespan and significant Stokes shift, LaAlO_3_ is widely used in the field of luminous materials [17,18,19,20]. In addition, numerous researchers have reported on its additional characteristics, such as poor heat conductivity [21,22].

Nevertheless, there is little research focusing on the optical characteristics of LaAlO_3_, particularly with regard to the materials used for reflective coatings materials. In order to consistently maintain a high and stable reflectivity coating material, it must retain its original crystal structure and not undergo a phase transition as the conditions change. The high symmetry of the crystal structure is advantageous for boosting the inter-band transition and has a significant impact on the optical characteristics of materials, according to the findings of earlier studies. Additionally, the intrinsic qualities of materials are also impacted by particle size homogeneity. Therefore, based on its high symmetry structure and thermal stability, LaAlO_3_ may be able to replace other inorganic compounds currently used as high-reflectance materials.

Unfortunately, the present research disregards the effects of the LaAlO_3_ perovskite preparation procedure [23,24,25]. For instance, a high sintering temperature results in irregular particles and rough grains, which restricts the optical characteristics [26,27]. Additionally, the LaAlO_3_ lattice will release oxygen atoms and create oxygen vacancies [8,28,29,30,31] at high temperatures, leading to a dramatic decrease in reflectivity, according to the phase transition mechanism. The band gap of LaAlO_3_ narrows as a result of the creation of oxygen vacancies and defect energy levels within the band gap [32,33,34,35,36]. Additionally, the oxygen vacancies will distort the perfect cubic perovskite lattice of LaAlO_3_ to some extent, lowering the material’s reflectivity. Exploring a technique to lessen the oxygen vacancy flaws in LaAlO_3_ is, therefore, significant [37,38,39,40]. The structure can also be tuned, and high-temperature flaws can be prevented by optimizing the preparation conditions based on the current preparation process.

Nb^5+^ doping has recently been discovered to be advantageous for enhancing the reflectivity of LaSrTiO_3_ materials [41]. The role and impact of Nb^5+^ doping in LaAlO_3_ perovskite, however, have not been explored. In this study, the high-temperature solid-state method was used to synthesize Nb^5+^-doped LaAlO_3_ perovskite material. This paper investigates the potential changes in microstructure, oxygen vacancy defects, and optical properties resulting from the partial replacement of Al^3+^ with Nb^5+^ in the LaAlO_3_ perovskite structure through a high-temperature solid-state reaction. Additionally, this paper examines the influence of oxygen vacancy on the band gap and reflectivity of materials. To the best of our knowledge, no previous research has been conducted on the impact of Nb^5+^-doped LaAlO_3_ on reflectivity.

## 2. Experimental

The high-temperature solid-state method was used to create LaNb*_x_*Al_1−*x*_O_3+δ_ perovskite powders. LaNb*_x_*Al_1−*x*_O_3+δ_ (*x* = 0, 0.05, 0.10, 0.15, 0.20, 0.20) was made from La_2_O_3_, Al_2_O_3_, and Nb_2_O_5_ for the high-temperature solid-state method. All raw materials are of analytical grade and sourced from Aladdin Reagent Company Limited. The high-temperature solid-state preparation technique is used conventionally. Raw materials are weighed according to the stoichiometric ratio, and anhydrous ethanol is added. The raw materials are then placed into a nylon ball milling tank. First, wet grinding and mixing are conducted in a ball mill, uniformly mixing the powders, drying, sieving with a 40-mesh sieve, and sintering at 1250 °C for 3 h in an air atmosphere to make samples. LaNb*_x_*Al_1−*x*_O_3+δ_ powder with a high reflectivity was produced in this manner.

X-ray diffractometry (XRD, D/max 2550V, RIGAKU, Tokyo, Japan) using Cu-K radiation (λ = 0.1542 nm) was used to observe the phase composition of the samples and the degree of lattice variation, and transmission electron microscopy (TEM, JEM-2100F, JEOL, Tokyo, Japan) was used to analyze the variation in the crystal plane d spacing and microscopic morphology. Scanning electron microscopy (SEM, Magellan 400, FEI, Hillsboro, OR, USA) was used to examine the microscopic morphology. Additionally, measurements of electron paramagnetic resonance (EPR, Bruker EMXplus-6/1, Herborn, Germany) were conducted to ascertain the number of oxygen vacancies. The photoluminescence spectra were recorded by a steady-state transient fluorescence spectrometer (PL, Fluorolog-3, HORIBA, Tokyo, Japan). An ultraviolet-visible-near-infrared spectrophotometer (UV-Vis-NIR, Lambda 1050, PerkinElmer, Waltham, MA, USA) was used to measure the reflectivity from 250 nm to 2500 nm. Samples were mixed proportionally with LaNb*_x_*Al_1−_*_x_*O_3+δ_ perovskite powder and 6% wt Polyvinyl Alcohol (PVA) binder, mixed evenly, and then transferred to a tablet mold. Under a pressure of 4 MPa for 2 min, a small disk with a diameter of 30 mm and a thickness of 2 mm was formed. The plate was placed in a muffle furnace and heated at 650 °C for 3 h to discharge the PVA binder.

## 3. Results and Discussion

The crystal structure of LaAlO_3_ is shown in Figure 1. The Vesta software (Ver:3.0.1) was used to expand the cell of the LaAlO_3_ structure by 2 × 2 × 2. The lowest formation energy Nb atom was then selected to replace the Al atom, resulting in the schematic diagram shown in Figure 1a. Figure 1b shows the XRD patterns of the LaNb*_x_*Al_1−_*_x_*O_3+δ_ (*x* = 0, 0.05, 0.10, 0.15, 0.20, 0.25) samples. The main diffraction peaks of LaNb*_x_*Al_1−_*_x_*O_3+δ_ were indexed to LaAlO_3_ with structure in accordance with the PDF card no. 85-0548. The diffraction peaks shifted slightly to a lower angle with an increase in the Nb^5+^ doping content from 0.05 mol% to 0.20 mol%, which could be attributed to the successful incorporation of Nb^5+^ into the LaAlO_3_ lattice. To obtain the accurate lattice parameters, the XRD patterns of LaNb*_x_*Al_1−_*_x_*O_3+δ_ were refined, and the refined results are shown in Table 1. Based on the structure, the lattice volumes of LaNb*_x_*Al_1−_*_x_*O_3+δ_ (*x* = 0, 0.05, 0.10, 0.15, 0.20, 0.25) were calculated to be 54.13, 54.16, 54.28, 54.49, 54.54, and 54.58 Å^3^, respectively. Obviously, the lattice volume of LaAlO_3_ increased from 54.13 to 54.58 Å^3^ with Nb^5+^ doping. The increase in lattice volume can be attributed to the partial substitution of Al^3+^ by Nb^5+^, whose Nb^5+^ has a slightly larger radius (0.78 Å) than Al^3+^ (0.675 Å) [42].

After doping, Nb^5+^ ions are dispersed into the lattice point of LaAlO_3_ perovskite structure, occupying Al^3+^ position, accompanied by a small number of heterogeneous diffraction peaks, which are consistent with the diffraction peaks of LaNbO_4_ (PDF#71-1405) and La_3_NbO_7_ (PDF#71-1345), indicating that most Nb^5+^ has been doped into the lattice of LaAlO_3_. It can be inferred that the reaction of synthesizing LaNb*_x_*Al_1−_*_x_*O_3+δ_ powder is shown in Formula (1): (1)$${\mathrm{La}}_{2}O_{3}+\left(1-x\right){\mathrm{Al}}_{2}O_{3}+x{\mathrm{Nb}}_{2}O_{5}\overset{\mathrm{heat}}{\to }2\mathrm{La}{\mathrm{Nb}}_{x}{\mathrm{Al}}_{1-x}O_{3+\delta }.$$

Under this reaction’s conditions, La_2_O_3_ and Nb_2_O_5_ can also react in a solid state at a high temperature, and the chemical reaction equation is shown in Formula (2):(2)$${2\mathrm{La}}_{2}O_{3}+{\mathrm{Nb}}_{2}O_{5}\overset{\mathrm{heat}}{\to }{\mathrm{LaNbO}}_{4}+{\mathrm{La}}_{3}{\mathrm{NbO}}_{7}.$$

Moreover, the detailed microstructure of the LaNb_x_Al_1−x_O_3+δ_ samples was investigated by high-resolution transmission electron microscopy (HR-TEM). As shown in Figure 1c, the grains of LaNb*_x_*Al_1−_*_x_*O_3+δ_ exhibit clear lattice fringes, and the d spacing of the Nb^5+^ doping sample is a little bigger than that of the (110) plane of LaAlO_3_. With the increase in doping content, the interplanar spacing of 110 crystal faces increased from the initial 0.379 Å to 0.404 Å. The lattice spacing change observed by HR-TEM is consistent with a shift toward lower angles in the XRD patterns. In the case of 0.25 mol% doping, the XRD diffraction peaks shift toward higher angles, most likely due to the significant growth trend of LaNbO_4_ under 0.25 mol% Nb^5+^ doping, causing changes in atomic positions within the crystal, leading to lattice non-uniformity and alterations in the crystal structure, thus resulting in the shift of XRD diffraction peaks toward higher angles. At this point, the interplanar spacing observed in TEM also shows irregularities.

The SEM surface image of LaNb*_x_*Al_1−*x*_O_3+δ_ is shown in Figure 2. Based on the SEM image, it can be concluded that the growth of the LaAlO_3_ perovskite is of high quality, and the growth steps are clearly visible. This suggests that the LaAlO_3_ material prepared has excellent crystallinity and lattice integrity, meeting the requirements for a perovskite structure. This is of great significance for the performance and application of the material. At high magnification (30,000×), for the Nb^5+^ doping level between 0.10 mol% and 0.20 mol%, the structure of the crystal particles shows the complete morphology. Perovskite structure growth stages show the clearest crystal structure at a doping concentration of 0.15 mol%, the rough bulk structure disappears, and the perovskite structure becomes more pronounced due to the role played by Nb^5+^ as a sintering aid. The lamellar structure appears when the doping content exceeds 0.25 mol%. As can be seen, this sample has a distinct grain boundary. 

Drawing the particle size distribution curve of 100 particles in the image reveals that (a) the particle size of Nb^5+^ undoped LaAlO_3_ is mostly in the range of 0.4–1.0 μm; (b) the particle size of 0.05 mol% Nb^5+^ doped LaAlO_3_ is in the range of 0.4–0.8 μm; (c) the particle size of 0.10 mol% doped LaAlO_3_ is in the range of 0.4–0.8 μm; (d) the particle size 0.15 mol% doped LaAlO_3_ is in the range of 0.2–0.8 μm; (e) the particle size of 0.20 mol% doped LaAlO_3_ is in the range of 0.4–1.2 μm; and (f) the particle size of 0.25 mol% doped LaAlO_3_ is in the range of 0.2–1.2 μm. 

Figure 3 shows the lamellar structures in LaNb_0.25_Al_0.75_O_3+δ_. The appearance of lamellar structures resulting from 0.25 mol% Nb^5+^ doping leads to the material’s unevenness and changes in the crystal structure. This result is consistent with the conclusions obtained by SEM.

The EPR spectrum of the complex form of LaNb*_x_*Al_1−*x*_O_3+δ_ is provided by the anisotropy of the EPR signal and the superposition of concentrated EPR signals from distinct defect types, as shown in Figure 4a. EPR technology is a common means to quantitatively characterize the concentration of oxygen vacancy defects. By measuring and analyzing the standard sample with known concentration, the quantitative relationship between the concentration of oxygen vacancy defects and the EPR signal can be established so as to realize the quantitative characterization of the concentration of oxygen vacancy defects in the sample. When unpaired electrons interact with an external magnetic field, energy level transition, and resonance absorption will occur. In solid materials, oxygen vacancies usually contain unpaired electrons, which will participate in the EPR process. Due to the influence of the electronic environment around oxygen vacancy, the EPR signal that leads to oxygen vacancy defect usually has a characteristic peak near g ≈ 2.000. This characteristic peak corresponds to the resonance absorption of unpaired electrons. In order to reduce the experimental error, the same test conditions and atmosphere are used for the test. It is evident that there is a large EPR signal peak at the peak position of g ≈ 2.000, where hv = gβB predicts that the known sample’s oxygen vacancy concentration is g ≈ 2.000. Where h is Planck constant, ν is microwave frequency; β is Bohr magneton; B is magnetic induction intensity, and g is the g-factor. Through testing the EPR signal peaks of oxygen vacancies at 300 K (Figure 4a), it can be observed that with an increasing Nb^5+^ doping concentration, the oxygen vFacancy concentration decreases. Figure 4a shows the specific quantitative data. It is evident that the oxygen vacancy concentration decreases significantly with increasing doping content. The vacancy of undoped oxygen is reduced from 1.01613 spins/g to 0.35556 spins/g. But when the doping content reached 0.20 mol% and 0.25 mol%, the oxygen vacancy concentration began to increase. This may be due to the structural change caused by high-concentration doping and the growth of the second phase. To further confirm the defect concentration of LaNb*_x_*Al_1−*x*_O_3+δ_ material, photoluminescence (PL) spectroscopy was performed in Figure 4b. As can be seen from this Figure, there are five luminescence peaks, which are 482 nm double-ionized oxygen ($V_{O}^{**}$), 530 nm monoionized oxygen vacancy ($V_{O}^{*}$) and oxygen vacancy at 565 nm, respectively [43]. With the appearance of a large number of carriers, some monoionized oxygen vacancies and double-ionized oxygen vacancies are transformed into oxygen vacancies. In addition, the luminescence peaks at 492 nm and 508 nm are mainly caused by the change in energy band structure caused by lattice defects, and the reason for this needs further study. Therefore, the main optical absorption centers in LaNb*_x_*Al_1−*x*_O_3+δ_ materials are oxygen vacancy defects and lattice defects.

The reflectance spectra are used to characterize the optical characteristics of LaNb*_x_*Al_1−*x*_O_3+δ_ samples. The reflectance spectrum of the samples prepared with various Nb^5+^ doping levels by the high-temperature solid-state method is shown in Figure 5a. Reflectivity increases initially and then decreases as Nb^5+^ content increases. The reflectivity is the greatest at 0.15 mol% of doping. The greatest reflectivity reaches 99.7% at 1050 nm in the near-infrared light band with LaNb_0.15_Al_0.85_O_3+δ_, while LaAlO_3_ is only 95.8%. 

The improvement in reflectivity is significantly influenced by the changes in phase composition and material structure. According to the SEM image in Figure 2, it is known that when the Nb^5+^-doped concentration rises to 0.15 mol% and the reflection path and interface number expand after doping. Combining the XRD and SEM analysis of the materials, it is found that with the doping content of 0.20 mol% and 0.25 mol%, the impurity peaks are obviously increased, and other phase structure particles appear, so there are more components affecting the energy band structure, and reflectivity changes greatly.

According to the UV-Vis diffuse reflectance spectroscopy in Figure 5a. All of the samples exhibit a distinct absorption edge, and the absorption edges are all located near the wavelength of 310 nm. Based on the UV-Vis DRS spectra, the band gap energy (Eg) of the samples was estimated via Kubelka–Munk as follows [44,45]:(3)$${\left(\alpha h\nu \right)}^{n}=A(h\nu -\mathrm{Eg})$$
where h is the Planck constant; ν is the photon frequency; α is the absorption coefficient, and A is the proportional constant. The value of the index n is determined by the properties of the sample transition: for direct band gap semiconductors, n = 0.5; for indirect band gap semiconductors, n = 2 [46]. Considering the nature of optical transition in LaAlO_3_ as an indirect transition [47], the band gaps for the respective LaNb*_x_*Al_1−*x*_O_3+δ_ composites were evaluated, as shown in Figure 5b,c. Obviously, with the increase in Nb^5+^ content, the band gap of LaNb_x_Al_1−x_O_3+δ_ prepared by the solid-state method first increases and then decreases. 

The effect of oxygen vacancy on the optical properties of LaAlO_3_ materials is significant. Burstein–Moss effect proposed by the Pauling incompatibility principle emphasizes that when additional energy levels are added between VB and CB, the band gap of the material narrows, allowing for electrons to absorb lower energy, thus realizing the transition behavior. The more electrons in the energy band, the stronger the ability to move from low energy level to high energy level. As shown in Figure 6, when light passes through the crystal, oxygen vacancies act as absorption centers in the crystal, resulting in a decrease in reflectivity because oxygen vacancies generate oxygen vacancy levels in the band gap interval. Electrons absorb energy and make it jump from the valence band to the conduction band. In addition, the generation of oxygen vacancies will also lead to changes in the electronic structure of materials, which will lead to changes in their optical properties. When Nb^5+^ is added into the LaAlO_3_ lattice, the defect equation can be used to describe the doping mechanism, which proves that Nb^5+^ inhibits oxygen vacancies, neutralizes oxygen vacancy defects, and, therefore, widens the band gap, decreases the ability of electrons to absorb energy and transition to the conduction band, and boosts reflection while decreasing absorption. Combined with Figure 4, it is not difficult to see that with the increase in Nb^5+^ doping content, the oxygen vacancy concentration decreases. The reflectivity is influenced by multiple factors, such as microstructure defects and oxygen vacancy defects. Because there are more microstructure defects in 0.20 mol% and 0.25 mol% doping, the reflectivity and band gap are obviously lower than that in 0.15 mol% doping.

## 4. Conclusions

In this study, powders of LaNb*_x_*Al_1−*x*_O_3+δ_ with high reflectivity were prepared by a high-temperature solid phase reaction method. Evidence shows that Nb^5+^ successfully entered the LaAlO_3_ lattice, inducing changes in the lattice constants and crystal plane spacing. The lattice volume increases with increasing Nb^5+^ doping, which is due to the fact that the radius of Nb^5+^ is slightly larger than that of Al^3+^. The sintering-assisted effect of Nb^5+^ leads to a gradual reduction in microstructural defects in the particles; the powder of LaNb_0.15_Al_0.85_O_3+δ_ tends to range in size from 0.2 to 0.8 μm. Based on the quantitative study of EPR and PL emission peaks, it is clearly seen that the concentration of oxygen vacancies decreases with increasing doping, which proves the important role of Nb^5+^ in neutralizing the charge during high-temperature sintering.

Fewer microstructural defects contribute to the enhancement of the light reflection path, and lower oxygen vacancy concentration contributes to fewer oxygen vacancy defect energy levels and increases the band gap, which reduces the ability of the material to absorb photon jumps. The superposition of the two effects contributes to further enhancing the reflectivity of the material. For LaNb*_x_*Al_1−*x*_O_3+δ_, the maximum reflectivity reaches 99.7%, which appears at 1050 nm doped with 0.15 mol% Nb^5+^, while the pristine LaAlO_3_ sample can only reach 95.8%. This study presents a feasible method to reduce sintering defects and improve the optical properties of LaAlO_3_ material. This has the potential to create a new category of highly reflective materials.

## Figures and Tables

**Figure 1 nanomaterials-14-00608-f001:**
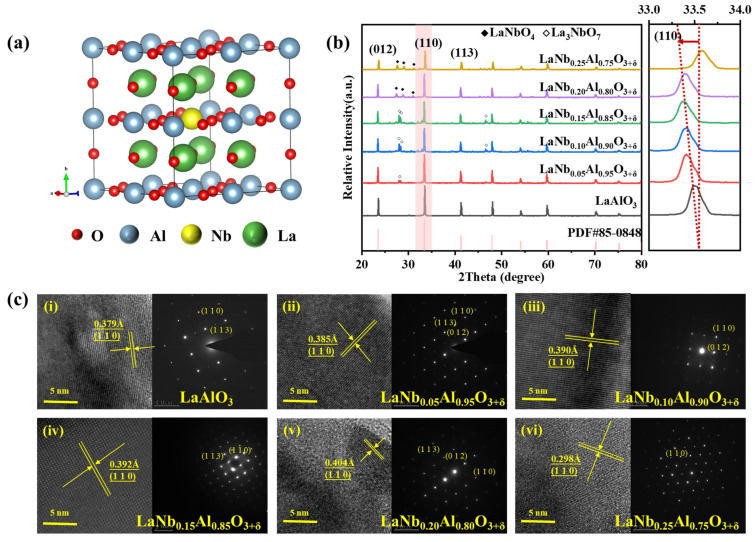
(**a**) Schematic illustration of 2 × 2 × 2 supercell of LaNb*_x_*Al_1−*x*_O_3+δ_ perovskite crystal. (**b**) XRD patterns for pure (LaAlO_3_) and Nb^5+^ doped LaAlO_3_ with various Nb: LaNb*_x_*Al_1−*x*_O_3+δ_ by high-temperature of solid-state method. Enlarge image of LaNb*_x_*Al_1−*x*_O_3+δ_ (*x* = 0.05–0.25) ceramics at 2θ range of 33~34°. The red dotted line represents the shift trend of the diffraction peak of the (110) crystal plane. (**c**) TEM images of LaNb*_x_*Al_1−*x*_O_3+δ_ synthesized by high-temperature solid-state method. (i) A total of 0 mol% Nb^5+^-doped LaAlO_3_. (ii) A total of 0.05 mol% Nb^5+^-doped LaAlO_3_. (iii) A total of 0.10 mol% Nb^5+^-doped LaAlO_3_. (iv) A total of 0.15 mol% Nb^5+^-doped LaAlO_3_. (v) A total of 0.20 mol% Nb^5+^-doped LaAlO_3_. (vi) A total of 0.25 mol% Nb^5+^-doped LaAlO_3_.

**Figure 2 nanomaterials-14-00608-f002:**
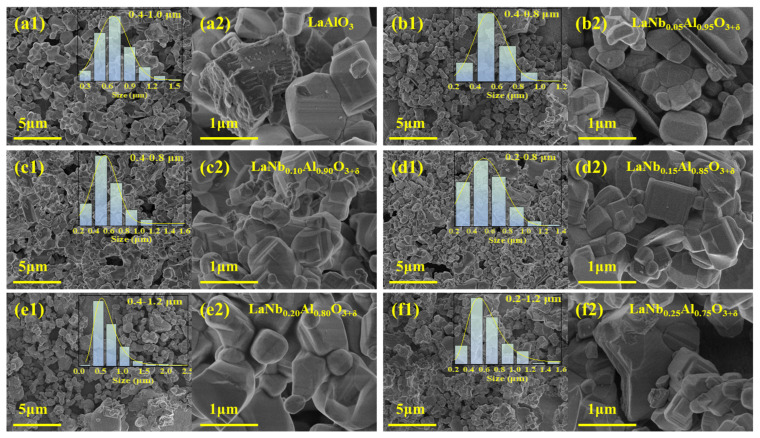
Grain growth observed by SEM on LaNb*_x_*Al_1−*x*_O_3+δ_ specimens formed by doping with different Nb^5+^ contents and particle size distribution curves. (**a**) A total of 0 mol% Nb^5+^-doped LaAlO_3_. (**b**) A total of 0.05 mol% Nb^5+^-doped LaAlO_3_. (**c**) A total of 0.10 mol% Nb^5+^-doped LaAlO_3_. (**d**) A total of 0.15 mol% Nb^5+^-doped LaAlO_3_. (**e**) A total 0.20 mol% Nb^5+^-doped LaAlO_3_. (**f**) A total 0.25 mol% Nb^5+^-doped LaAlO_3_. Within each content group, the magnifications of the images were 5000× and 30,000×, respectively.

**Figure 3 nanomaterials-14-00608-f003:**
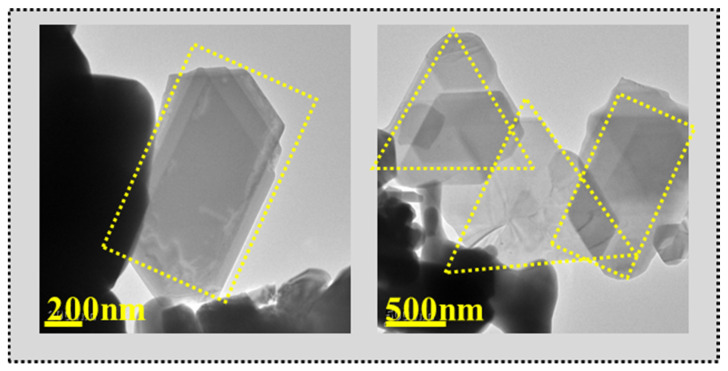
LaNb_0.25_Al_0.75_O_3+δ_ sample lamellar structure in TEM images.

**Figure 4 nanomaterials-14-00608-f004:**
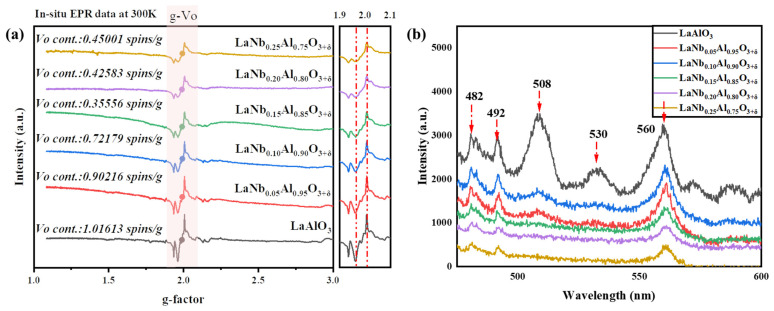
(**a**) In situ EPR data at 300 K. (**b**) PL spectra of LaNb*_x_*Al_1−*x*_O_3+δ_ at different doping ratios. The red arrow points defect luminescence peak.

**Figure 5 nanomaterials-14-00608-f005:**
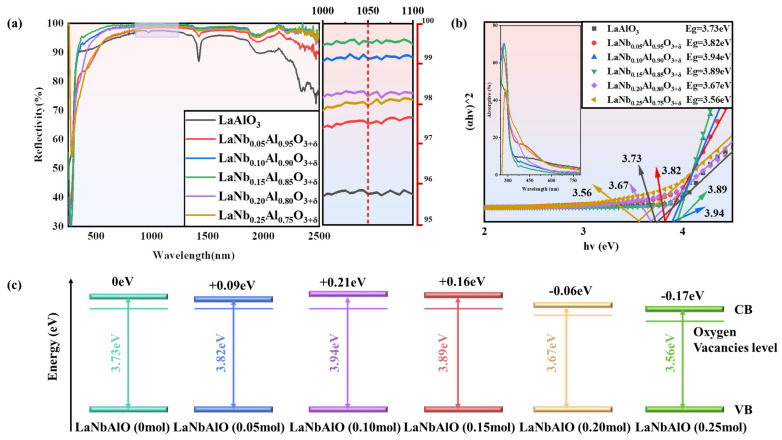
(**a**) Reflectance spectra of LaAlO_3_ samples doped with different contents of Nb^5+^. The enlarged image has a wavelength of 1000–1100 nm. (**b**) Optical band gap obtained by diffuse reflection spectrum using Kubelka–Munk equation. (**c**) Schematic illustration of band structure of LaNb*_x_*Al_1−*x*_O_3+δ_. (From left to right, the doping content is 0, 0.05, 0.10, 0.15, 0.20, 0.25 mol%).

**Figure 6 nanomaterials-14-00608-f006:**
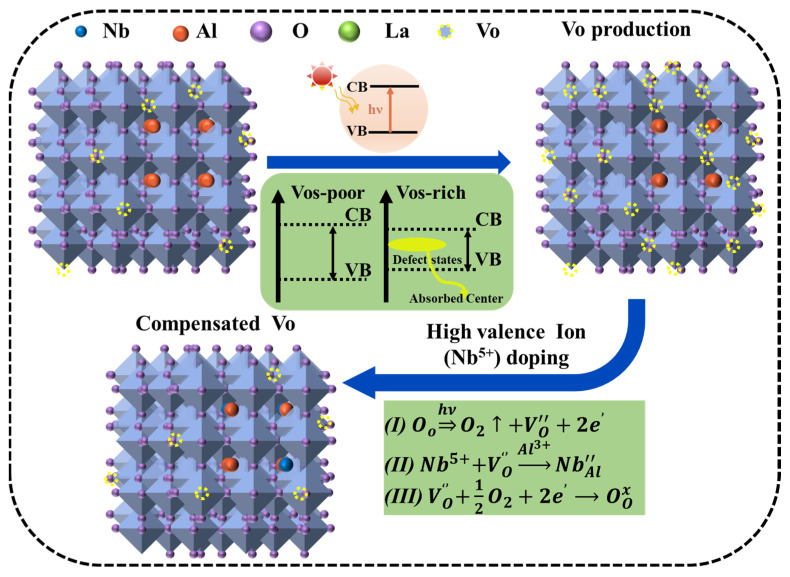
Schematic illustration of vacancy migration of LaNb*_x_*Al_1−*x*_O_3+δ_ under light conditions.

**Table 1 nanomaterials-14-00608-t001:** Refined results for the LaNb*_x_*Al_1−*x*_O_3+δ_ (*x* = 0, 0.05, 0.10, 0.15, 0.20, 0.25) samples.

	Lattice Parameters (Refined)						
	a (Å)	b (Å)	c (Å)	α (deg)	β (deg)	γ (deg)	V (Å^3^)
*x* = 0	3.78274	3.78274	3.78274	89.9629	89.9629	89.9629	54.13
*x* = 0.05	3.78338	3.78338	3.78338	89.9568	89.9568	89.9568	54.16
*x* = 0.10	3.78628	3.78628	3.78628	89.9597	89.9597	89.9597	54.28
*x* = 0.15	3.79118	3.79118	3.79118	89.9788	89.9788	89.9788	54.49
*x* = 0.20	3.79239	3.79239	3.79239	89.9566	89.9566	89.9566	54.54
*x* = 0.25	3.79321	3.79321	3.79321	89.9814	89.9814	89.9814	54.58

## Data Availability

Data are contained within the article.

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
