# Peer review of "Optimizing the Structure and Optical Properties of Lanthanum Aluminate Perovskite through Nb5+ Doping"

_nanomaterials, 2024, doi:10.3390/nano14070608_

Round 1

Reviewer 1 Report (Previous Reviewer 2)

Comments and Suggestions for Authors

Dear authors,

Thank you for the corrections made according to my suggestions.

The article is coherent and easy to read.

Best Regards

Author Response

Dear Professor,

Thank you for your kind words and feedback. We appreciate your time and effort in reviewing our article. Your suggestions were invaluable in improving the coherence and readability of the manuscript.

We are grateful for your support and guidance throughout this process.

Best regards,

Wei Liu

Reviewer 2 Report (New Reviewer)

Comments and Suggestions for Authors

The authors, Wei Liu and co-workers, did an interesting study optimizing the structure and optical properties of lanthanum aluminate perovskite material doped with the rare earth ion Nb5+. They did XRD, SEM, TEM, EPR, and reflectance measurements, in order to give a better insight in synthesized samples (obtained by solid-state reaction).

Although I think that the paper should be accepted for publication, I have some remarks for improvement.

First, an English correction must be done. When we talk about doping, it is always accompanied by %, at%, mol %. 

1) In the introduction part, the authors must emphasize the novelty of their work, as well as the application of the investigated material.

2) The use of appropriate references throughout the paper is mandatory, as well as a detail literature overview. 

The discussion in the XRD section is not well done. It is a bit confusing. In Fig. 1b, you presented additional phases, but they are not explained well. Also, we see a change in peak position, but you should give a table with the crystallite size, cell parameters, and so on. It would be easier to follow. 

Why didn't you try to obtain the pure structure, maybe by using different temperatures of sintering, or a longer/shorter time? Did you consider that? It would be definitely better for future applications. 

TEM figures are quite small. SAED can't be seen properly, as well as the planes that you calculated. 

The authors should include some luminescent measurements in the paper, to present a potential use for this material. 

The conclusion should be improved in order to emphasize the achieved results.

Comments on the Quality of English Language

English correction must be done. 

Author Response

Dear Professor,

Thank you for carefully reviewing the papers submitted by us and putting forward valuable suggestions for improvement. We are very grateful for your guidance and advice. In response to your suggestions for revision, we have conducted detailed discussion and analysis, and made the following amendments:

Your questions:

  1. First, an English correction must be done. When we talk about doping, it is always accompanied by %, at%, mol %.

    Thank you for your suggestion, professor, I have revised for the expression of mol involved in the article, the mol in the whole text are revised to the expression of mol%, thank you for your suggestion of modification

  1. In the introduction part, the authors must emphasize the novelty of their work, as well as the application of the investigated material

    Yes, Professor, thank you for your suggestion. The material is innovative in two ways. (1). The doping of Nb5+ helps sintering and promotes grain growth, thus suppressing grain microstructural defects. (2). Doping of Nb5+ promotes charge neutralisation and thus suppresses oxygen vacancy defects. These studies have not been reported in the literature to a larger extent before, while there are almost no studies on Nb5+ doped LaAlO3 materials in terms of high reflectivity coating materials. These have now been added to the last paragraph in the INTRODUCTION, thank you for your comments.

  1. The use of appropriate references throughout the paper is mandatory, as well as a detail literature overview

    Thank you for your opinion. At present, the references have been supplemented and corrected according to your opinion.

  1. The discussion in the XRD section is not well done. It is a bit confusing. In Fig. 1b, you presented additional phases, but they are not explained well. Also, we see a change in peak position, but you should give a table with the crystallite size, cell parameters, and so on. It would be easier to follow

    Thank you for your opinion. According to your proposal, the lattice parameters and lattice volume before and after doping have been calculated. The results have been added to Table 1. Thank you for your suggestion, which will help us to better understand the influence of doping on the crystal structure of LaAlO3.

  1. Why didn't you try to obtain the pure structure, maybe by using different temperatures of sintering, or a longer/shorter time? Did you consider that? It would be definitely better for future applications.

    In fact, we have carried out experiments over a wide range of temperature intervals and sintering durations, such as sintering at 1100°C and 1200°C for 3h and 6h, 1300°C, 1400°C, 1500°C for 3h and 6h, respectively. However, the LaAlO3 phase was not fully generated under sintering at 1100 and 1200°C, and at 1300°C, 1400°C and 1500°C, the It was observed that the XRD diffraction peaks only changed in intensity and had no effect on the LaNbO4 impurity peaks. In addition, the sintering phenomenon becomes more pronounced due to high temperatures, resulting in larger particle sizes. In addition, higher temperatures lead to the formation of more oxygen vacancies. These affects the reflectivity of the material. Considering all these factors, we chose to prepare the material at 1250°C for 3 hours.

  1. TEM figures are quite small. SAED can't be seen properly, as well as the planes that you calculated

    Thank you for your suggestion. According to the issues you mentioned about the small size of TEM images and the unclear SAED patterns, adjustments have been made to focus on presenting the electron diffraction and interplanar spacing more clearly in the images. This will facilitate a better understanding of the key information by the readers.

  1. The authors should include some luminescent measurements in the paper, to present a potential use for this material?

    I'm sorry, professor. It may have caused a misunderstanding in our research because we used PL spectroscopy to characterize oxygen vacancies in the test. The main reason for using PL spectroscopy in the experiment is to study the existence of luminescent peaks related to oxygen vacancies and the variation in luminescence intensity. This is quite different from the common understanding of luminescent materials. Our application field is mainly focused on high reflectivity coating materials, which are not closely related to the application of luminescent materials. Although LaAlO3 material has structural advantages that may have some applications in luminescent materials, for the Nb5+ doped LaAlO3 we are currently testing, it is not sufficient for use as a luminescent material. If there is an opportunity in the future, additional elemental doping can be considered to achieve its application in the field of luminescent materials.

  1. The conclusion should be improved in order to emphasize the achieved results

    Thank you for your advice. It is true that there was a lack of conclusive statements, and we have now made corrections to the conclusion section and emphasized the results better. This helps to improve the accuracy and credibility of the research. Thank you for your feedback; it is crucial for continuously refining and amending the conclusiveness of the research.

    We will take every suggestion you put forward seriously and strive to ensure the improvement of the quality of the paper. Thank you again for your guidance and attention, and look forward to receiving your further feedback on our revised paper.

Good wishes!

Reviewer 3 Report (New Reviewer)

Comments and Suggestions for Authors

The paper describes selected properties of LaAlO3 system doped with Nb5+ ions. The highlighted text would indicate that the manuscript has been already revised; nevertheless, in my opinion, it still cannot be accepted for publication. The main reasons are insufficient samples characterization and discussion of the results, which do not allow certain conclusions to be drawn, as well as many imprecise statements presented by the Authors. My detailed comments are included in the pdf file.

Comments on the Quality of English Language

English and errors should be corrected, for example, in these sentences/phrases: 

- Exploring a technique to lessen the oxygen vacancy flaws in LaAlO3 is therefore significance.

- LaNbxAl1-xO3+ (x=0, 0.05, 0.10, 0.15, 0.20, 0.20).

- Scann electron microscopy

- photoluminescence spectra was

- LaAlO3:Nb(x;x=0-0.25mol) 

- At high magnification (1μm)

- … photoluminescence (PL) spectroscopy was performed in Figure 3(b).

- According to the UV-Vis diffuse reflectance spectroscopy in Figure 4a.

- there are two 3-3 formulas in the text

Author Response

Dear Professor,

Thank you very much for your guidance, Professor. It is true that the manuscript may not meet your requirements, and I am trying my best to make revisions to minimize the errors and issues in the manuscript. I have carefully considered and made corrections based on your feedback from last time as well as this time. I appreciate your thorough review of my manuscript, and I deeply feel your dedication to this work. I believe you also hope to salvage my manuscript to meet the publication standards. I am working hard to revise and proofread, aiming to reduce colloquialisms and errors in expression. Your assistance and dedication are valuable assets and experiences for me. Once again, I express my sincerest gratitude to you and hope you can forgive my oversights and negligence. In response to your suggestions for revision, we have conducted detailed discussion and analysis, and made the following amendments:

Your questions:

  1. “LaAlO3:Nb (LaNbxAl1-xO3+δ) samples with niobium oxide”–thereis no Nb2O5 phase in the system

    Thank you for your comments, due to my negligence, here should be written Nb5+ doping, wrongly expressed as niobium oxide, has been corrected in the original text, I hope to be able to get your forgiveness.

  1. In the abstract, there is no information about impurity phases in the doped system

    Yes, professor, the reason for not providing excessive description of the impurity peaks in the abstract is twofold. On one hand, it is influenced by the word limit of the abstract section, and on the other hand, although LaNbO4 does exist in the system, it is not a primary factor affecting defects or reflectance. Therefore, in the abstract, to ensure the communication of the core content and key points of the research, the focus is solely on describing the main defects and reflectance factors. I hope to have your understanding.

  1. “Oxygen vacancy defects in different compositions are analyzed at different temperatures…” there is no analysis at different temperatures.

    I'm very sorry, professor. This is my negligence and mistake. During the previous modification process, I analyzed the oxygen vacancy concentration at different temperatures (100K, 300K, 500K) and found that the overall trend was consistent. Therefore, I overlooked this part and only described the oxygen vacancy concentration at 300K to better analyze, assess, and understand the data. I apologize for neglecting to modify the text during the chart revision.

  1. “This study demonstrates that Nb-doped LaAlO3 powder has broad application potential”–the broad application potential was not shown in the paper.

    Thank you for your question, this is the first step in our research, the research of high reflectivity powder with low defects, mainly for the application of high reflectivity coating materials, this is our main application prospects, about this part of the content of the article has also been added, thank you for your question!

  1. “X-ray diffraction machine (…) and transmission electron microscopy (…) are used to observe the microscopic morphology of a sample.”–XRD is not used to observe morphology.

    I am sorry Professor, here is indeed my expression of the error, XRD and TEM are two different characterisation methods, XRD is mainly applied to characterise the material phase composition, TEM is mainly applied to characterise the crystal spacing and microscopic morphology, in the article Experimental has been corrected, thank you for reminding.

  1. SEM “was used to examine the structure and element distribution of the material.”–distribution of the elements is not presented; “structure” is not precise in this sentence

    Thank you very much for pointing out this problem, EDS can be used to characterise the distribution of elements in the material, in the test of this paper, due to the effect of material charge, therefore EDS characterisation was not carried out, therefore for this sentence, the characterisation of "element distribution of the material" has been deleted, thank you for your valuable suggestion and understanding. Also, the "structure" you mentioned has been changed to "micromorphology" as requested.

  1. “Nb2O5 was used as the burning aid to guide the growth of LaAlO3”–what do you mean by naming Nb2O5“burning aid”?

    I'm sorry, Professor, here is my expression mistake. In this paper, Nb2O5 not only neutralizes charges but also acts as a sintering aid, promoting the sintering process, leading to a more uniform particle size distribution, more complete grain growth, and clearer grain boundaries. The term "burning aid" that I used is indeed not sufficient to clearly indicate this function, so I will modify it to "sintering aid".

  1. The whole paragraph describing XRD results should be re-written. It should be clearly stated that impurity phases were present in all doped systems. More detailed analysis should be performed to show among others the grain sizes of the powders, lattice parameters, and the phase composition. The description of Nb influence on lattice parameters is not clear. Can the radius of Nb5+ be called similar to Al3+ if the difference is 0.1A? Why position of diffractions for 0.25 sample is not following the trend (decreasing 2 theta angle)?

    Thank you for your opinion. According to your proposal, this section has been rewritten, the lattice parameters and lattice volume before and after doping have been calculated. The results have been added to Table 1. Thank you for your suggestion, which will help us to better understand the influence of doping on the crystal structure of LaAlO3. And we also reanalyzed the XRD spectrum according to the calculated lattice constants. There was a significant deviation in the diffraction peak for the 0.25mol doping. In addressing this issue, we can observe from the transmission electron microscope image in Figure 3 that under the 0.25mol doping condition, there is a noticeable increase in the number of lamellar structures. This leads to an increase in the non-uniformity of the crystal structure and the likelihood of greater distortion in the crystal structure, which may have caused the observed phenomenon.

  1. Looking at Figure 1, the samples doped with 0.1 or 0.15Nb seems to have different morphology from 0.05 or 0.2Nb. Are these images representative for the samples? Are the powders homogenous?

    Yes, professor, the samples with 0.10mol and 0.15mol doping did indeed exhibit this phenomenon, which may be due to the fact that these two samples were tested at different times from the other samples, resulting in differences in the test results. During the testing process, the samples themselves exhibited good uniformity, and we did not selectively choose regions with better crystallinity for testing. As for the samples with 0.05mol and 0.20mol doping, the surface morphology differs from the other samples, partly due to differences in grain size and partly due to variations in the selected regions. Therefore, we have modified the image for Figure 1 and conducted analysis solely on the TEM electron diffraction and interplanar spacing.

  1. Looking at results presented in Figure 2, there is no significant difference in samples morphology. Even the distribution of particles is quite similar. If the differenceis a result of different phases, it should be confirmed by TEM analysis showing different regions of the samples with different crystalline structures. Figure 2 caption: the figure do not show "grain growth"

    Thank you for your feedback. Based on your suggestion, we conducted a morphological analysis using transmission electron microscopy. Under the conditions of 0.25mol doping, a lamellar structure emerged. This lamellar structure may be the primary reason for the XRD diffraction peak shift. For other samples, during scanning electron microscopy testing, the samples were uniformly dispersed on copper foil, and samples from different regions exhibited a homogeneous state. Therefore, the SEM images can reflect the properties of the samples, although there are minor differences in microstructure. As the doping level increases, the rough large block samples do gradually decrease. Under 0.20mol and 0.25mol doping, a lamellar structure appeared in the SEM images, while the presence of the lamellar structure was more clearly observed in the Figure 3.

  1. “LaNbO4 has excellent reflectivity in the near ultraviolet band[42], according to the current investigation, which may be the primary cause of the improvement in near ultraviolet reflectivity.”–one may not see improvement in reflectivity in the UV range.

    Thank you for your suggestion, indeed, the role of LaNbO4 is not clear enough and there is less research on this part in this paper, therefore, after our deep consideration, we decided to delete this part of the expression, with your suggestion, the logical conception of the article is clearer, and a lot of unnecessary content is reduced, thank you very much for your help.

  1. The band gap widening after doping with Nb is not evident.

    We have conducted a thorough analysis of the issue you raised. By reanalyzing the curves and recalculating the band gap width using the Kubelka-Munk equation, we found that the overall change in the band gap width is relatively small. This change is primarily influenced by the good integrity and symmetry of the LaAlO3 crystal structure. In the process of preparing LaAlO3 using the high-temperature solid-phase method, the generation of oxygen vacancy defects is relatively low. Through Nb5+ doping, the oxygen vacancy concentration decreases, resulting in an initial increase in the band gap width. However, as the doping concentration increases, factors such as the appearance of lamellar structures lead to an increase in structural non-uniformity and the emergence of more defect energy levels, causing the band gap width to decrease. Thank you for your feedback. These further explanations contribute to a more comprehensive understanding of our research findings.

  1. The statements from the conclusion “With an increase in the amount of Nb5+ doping, the particles have a uniform cubic phase morphology” or “homogenous structure” are not confirmed by the results (impurity phases are present in the samples)

    Indeed, as you said, due to the existence of the second phase, the expression of "homogenous structure" mentioned in the study is not rigorous enough, and we sincerely accept your correction and suggestion to ignore the expression of its homogenous structure, and only focus on the reduction of the microstructural defects of the material after the doping of Nb5+, so that the rough structure in the system is gradually reduced, and thus the structural properties of the material are enhanced. In the conclusions, the characterisation on this part is all modified to a reduction of microstructural defects.

  1. What is the roughness of the samples(tablets)? Is it the same for all samples? This parameter is also important in reflectivity measurement.

    Yes, professor, surface roughness is indeed one of the factors affecting reflectance. In the process of powder reflectance testing, we carry out procedures such as grinding the samples, pre-treating with a 120-mesh sieve, and applying a pressure of 4MPa on a press for 2 minutes to ensure sample density. To avoid testing errors, all operations are conducted on the same day. Following your advice, we also aim to exclude the influence of surface roughness. However, we lack effective methods to test roughness for our plates. Therefore, we have conducted low-magnification scanning electron microscopy analysis on previously pressed samples, which exhibited smooth surfaces. These steps and analyses have eliminated the impact of roughness on the test results, ensuring the accuracy of the experimental results.

    Regarding the grammar and wording errors you pointed out in sections 15-25, they have been revised in the original text. Thank you very much for your corrections. My unclear expression has affected your reading experience, and I am truly grateful for your thorough review and inquiries. We will take every suggestion you put forward seriously and strive to ensure the improvement of the quality of the paper. Thank you again for your guidance and attention, and look forward to receiving your further feedback on our revised paper.

Good wishes!

Round 2

Reviewer 2 Report (New Reviewer)

Comments and Suggestions for Authors

After the revision the manuscript can be accepted for publication.

Author Response

Dear Professor,

Thank you for your kind words and feedback. We appreciate your time and effort in reviewing our article. Your suggestions were invaluable in improving the coherence and readability of the manuscript.

We are grateful for your support and guidance throughout this process.

Best regards,

Wei Liu

Reviewer 3 Report (New Reviewer)

Comments and Suggestions for Authors

I appreciate Authors' efforts put into improving the quality of their work. 

I have additional remarks (and some previous that have not been answered) presented in the attached file.

I would like to note that all changes  (even minor) made in the revised manuscript should be marked (which was not the case here).

Comments on the Quality of English Language

English still needs improvement

Author Response

Dear Professor,

    I hope this message finds you well. I wanted to express my gratitude for the valuable feedback you provided on my manuscript during our last meeting. I have made the revisions to the manuscript as your suggestions, and the modified content is included in the attached document. These adjustments are aimed at enhancing the accuracy and credibility of the research.

    I sincerely appreciate your patient guidance and insights, which are invaluable to the progress of my work. I look forward to your review of the revised manuscript and hope to receive your approval and further guidance.

    Thank you once again for your support and assistance.

Best regards,
Wei Liu

This manuscript is a resubmission of an earlier submission. The following is a list of the peer review reports and author responses from that submission.

Round 1

Reviewer 1 Report

Comments and Suggestions for Authors

The authors have prepared NbO+LAO polycrsytalline mixed samples by solid state reaction and studed their structure and optical properties. They claimed the formation of Nb-doped LAO, but I did not find a support for this. According to XRD the samples are rather mixed as they contain additional peaks, likely originated from LaNbO secondary phase. The reported marginal 2%-increase of the reflectance as well as mainly speculated changes of the band structure are not scientifically sound and do not merit publication in the Nanomaterials. My detailed comments are enclosed in the pdf file.

I do not recommend publication of this manuscript. May be it can be reconsidered in another journal after major revision.

Comments on the Quality of English Language

Quality of English is poor: one can find a lot mistakes.

Reviewer 2 Report

Comments and Suggestions for Authors

Dear authors

I have some questions and comments about the article.

My questions:

11  If the radius of niobium is larger than the radius of aluminum, then stress must arise; its value for individual niobium additions can be calculated, for example, from the Williamson-Hall expression.

22    Regarding Figure 1b, why do the LaNbO4 and La3NbO7 phases appear?

33 Why does peak 110 in Figure 1b stand out so much for the 0.25 mol composition?

44 We have the molar ratio of niobium. How much is the percentage of niobium actually incorporated into the LaAlO3 structure?

55 Figure 2 is not legible; the colors should be changed.

66 To the formula on line 182, you need to describe its symbols

77 Regarding the EPR results, what can be concluded about the aluminum atom's surroundings? Do we know from the EPR spectrum that Nb is substituted for lanthanum?

88 Is it possible to quantitatively calculate oxygen vacancies depending on the amount of niobium?

9 9 Figure 4b is difficult to read

110.  Can the authors determine the energy gap in these materials from other studies?

111.  the Figures require corrections and need to be unified

Reviewer 3 Report

Comments and Suggestions for Authors

The authors present paper on synthesis and characterization of LaAlO3:Nb nanoparticles. The topic is very interesting, the experimental part, synthesis and measurements, are clearly described and looks good. However, explanation of obtained results and conclusion are very unconvincing. The paper requires major revision at this stage.

1) p1.l30. "low dielectric constant" reference 4 states completely opposite in the second paragraph of Introduction. // optical band gap similar to that for semiconductors (from 5.6 to 7 eV) a high dielectric constant //

"good lattice matching" the specific material/application should be mentioned with which it matches well

2)  p1.l33 "Due to its strong optoelectronic capabilities, the B site held by Al3+ can be replaced by transition metal ions to produce a variety of applications, including solid electrolytes and high emissivity coatings." First and second part of that sentence don't make sense together.

3) p3.117 Please provide reference for ionic radii.

4) p3.l123 Your formula contains O3+δ. But in introduction you mentioned creation of oxygen vacancies at high temperatures. So is there oxygen excess or deficiency?  Another question is how the 5+ ion is stabilized in Al3+ position?

5) The formulas on Fig.1b are difficult to understand, especially LaNbAlO3(0mol). notation with 1-x / x is easier to read in my opinion. Also explain red dotted line in the right panel.

6) p4.l164 "... the bulk structure will decrease as ..." Something is missing in this sentence

7) p4.l168. "Obviously, due to the addition of Nb5+, the sintering is promoted, and the particle size tends to be uniform and refined" This sentence contradicts the data you mentioned just earlier. Increasing the doping the ranges of particle size are also increasing.

8) EPR signal comes from unpaired electrons. So to see it there must be F+ center (oxygen vacancy with one electron trapped), so only concentration of one type of vacancies can be evaluated. 

9) p5.l187 and Fig.3 Can you support your conclusions with some references or additional data. It does not look convincing at all. For example, 530nm signal, monoionized vacancy (as you state)is not visible in PL spectra in doped samples, while is present in EPR spectra with almost same signal intensity for x=0.05 and 0.10

10) p6.l206 " LaNbO4 has excellent reflectivity in the near ultraviolet band, according to the current investigation" Where does this conclusion come from?

11) Fig.4 The band gap for pure LaAlO3 is expected to be around 6 eV, you obtain 3.96 eV. Is it drastic change in fundamental band gap (Eg) or is it just defect absorption (not Eg)? If it is not Eg can you use it for Kubelka-Munk equation?

12) "Obviously, with the increase of Nb5+ content, the band gap of LaNbxAl1-xO3+δ prepared by solid-state method first increases and then decreases" First, this is totally not obvious, and the value you measured most likely is not a band gap, but defect levels in the gap.